# Specific Cerebrospinal Fluid SerpinA1 Isoform Pattern in Alzheimer’s Disease

**DOI:** 10.3390/ijms23136922

**Published:** 2022-06-22

**Authors:** Lorenzo Barba, Steffen Halbgebauer, Federico Paolini Paoletti, Giovanni Bellomo, Samir Abu-Rumeileh, Petra Steinacker, Federico Massa, Lucilla Parnetti, Markus Otto

**Affiliations:** 1Department of Neurology, Martin-Luther-University Halle-Wittenberg, 06120 Halle (Saale), Germany; lorbarba00@gmail.com (L.B.); samir.aburumeileh@gmail.com (S.A.-R.); petra.steinacker@uk-halle.de (P.S.); 2Department of Neurology, Ulm University Hospital, 89081 Ulm, Germany; steffen.halbgebauer@uni-ulm.de; 3Section of Neurology, Department of Medicine and Surgery, University of Perugia, 06132 Perugia, Italy; federico.paolinipaoletti@gmail.com (F.P.P.); giovanni.bellomo@unipg.it (G.B.); lucilla.parnetti@unipg.it (L.P.); 4Department of Neuroscience, Rehabilitation, Ophthalmology, Genetics, Maternal and Child Health, University of Genoa, 16132 Genoa, Italy; fedemassa88@gmail.com

**Keywords:** serpinA1, cerebrospinal fluid, biomarker, CIEF immunoassay, Alzheimer’s disease

## Abstract

SerpinA1 (α1-antitrypsin) is a soluble glycoprotein, the cerebrospinal fluid (CSF) isoforms of which showed disease-specific changes in neurodegenerative disorders that are still unexplored in Alz-heimer’s disease (AD). By means of capillary isoelectric focusing immunoassay, we investigated six serpinA1 isoforms in CSF samples of controls (*n* = 29), AD-MCI (*n* = 29), AD-dem (*n* = 26) and Lewy body disease (LBD, *n* = 59) patients and correlated the findings with CSF AD core biomarkers (Aβ42/40 ratio, p-tau, t-tau). Four CSF serpinA1 isoforms were differently expressed in AD patients compared to controls and LBD patients, especially isoforms 2 and 4. AD-specific changes were found since the MCI stage and significantly correlated with decreased Aβ42/40 (*p* < 0.05) and in-creased p-tau and t-tau levels in CSF (*p* < 0.001). Analysis of serpinA1 isoform provided good di-agnostic accuracy in discriminating AD patients versus controls (AUC = 0.80) and versus LBD patients (AUC = 0.92), with best results in patients in the dementia stage (AUC = 0.97). SerpinA1 isoform expression is altered in AD patients, suggesting a common, albeit disease-specific, in-volvement of serpinA1 in most neurodegenerative disorders.

## 1. Introduction

SerpinA1 (also known as α1-antitrypsin) is a soluble glycoprotein and, together with neuroserpin and α1-antichymotrypsin, belongs to the serpin superfamily [1]. It is mainly produced by hepatocytes, especially during acute inflammation, and acts as a serin-protease inhibitor with relevant anti-inflammatory, anti-infective and tissue-protective properties [1]. In the central nervous system, serpinA1 is associated with neuroinflammation and microglial activation [2,3]. An upregulation of the protein, as well as its deposition in amyloid-β (Aβ) and tau aggregates, was observed in the brains of patients affected by Alzheimer’s disease (AD) and frontotemporal lobar degeneration (FTLD) [4,5,6]. Hence, serpins might play a role in the pathophysiology of neurodegenerative disorders, particularly by taking part in protein misfolding [6,7,8] and in the regulation of neuroinflammation [2,4].

SerpinA1 can be quantified in cerebrospinal fluid (CSF), although the source of CSF serpinA1 is still not clear [7]. Changes in its absolute CSF concentrations were reported in Parkinson’s disease (PD), dementia with Lewy bodies (DLB) and AD [7,8]. Moreover, the qualitative assessment of serpinA1 has gained increasing attention in the last few years, as different expression patterns of serpinA1 isoforms have been described in several neurodegenerative disorders. The first report on PD with dementia (PDD) described a shift towards more acidic pH values and the presence of an additional acidic isoform in comparison to PD without dementia [9]. Subsequently, the same approach was applied to investigate patients with Creutzfeldt-Jakob disease (CJD) and frontotemporal lobar degeneration (FTLD), who showed peculiar patterns with higher expression of acidic and basic isoforms, respectively [10]. To date, no study has focused on serpinA1 isoforms and their relevance in AD. SerpinA1 isoforms reflect different post-translational modifications (PTMs) of the protein, which might be related to the neurodegenerative process in a disease-specific fashion. As an example, the development of dementia in PD was linked to hypersialylated serpinA1 [7]. Therefore, a deeper understanding of the mechanisms underlying such PTMs may suggest new therapeutic strategies for neurodegenerative disorders.

Hence, we took advantage of our previously described capillary isoelectric focusing (CIEF) immunoassay [9] to investigate qualitative modifications of CSF serpinA1 in AD, Lewy body disorders (LBDs, including PD and DLB) and control subjects. Our primary aim was to assess the relative distribution of serpinA1 isoforms in AD and the diagnostic accuracy of serpinA1 analysis to distinguish AD from other conditions. Secondly, we focused on AD patients in different disease stages, namely mild cognitive impairment (AD-MCI) and dementia (AD-dem), in order to evaluate whether serpinA1 expression differs along the clinical continuum.

## 2. Methods

### 2.1. Patients Selection

CSF samples were retrospectively selected from patients diagnosed with AD and LBD referring to the Section of Neurology in Perugia (University of Perugia, Perugia, Italy) from 2016 to 2020. As a part of the routine diagnostic workup, all patients underwent an extensive neuropsychological evaluation, including standard cognitive assessment with the mini-mental state examination (MMSE) [11], the Montreal Cognitive Assessment (MoCA) [12] and clinical dementia rating (CDR) [13], as well as brain neuroimaging (magnetic resonance imaging or computed tomography) and lumbar puncture.

AD was diagnosed according to the CSF profile (A+/T+), in line with the most updated National Institute on Ageing and Alzheimer’s Association (NIA-AA) recommendations [14]. Based on the CDR score, AD patients (*n* = 55) were clinically stratified as affected by mild cognitive impairment (AD-MCI, *n* = 29, CDR = 0.5) and dementia (AD-dem, *n* = 26, CDR ≥ 1.0) [15,16].

The LBD group (*n* = 59) included patients with diagnoses of PD (*n* = 52) and DLB (*n* = 7) according to the currently available criteria [17,18]. LBD patients underwent an extensive neuropsychological evaluation, and patients within the PD subgroup were further classified as having mild cognitive impairment (PD-MCI, *n* = 19) or dementia (PDD, *n* = 12) according to the available criteria [19,20].

The control group (*n* = 29) was composed of subjects referring to the same Section of Neurology (University of Perugia, Perugia, Italy) for subjective memory complaint, who did not fulfill criteria for MCI [21] in the neuropsychological evaluation (*n* = 12), and for non-degenerative neurological conditions (1 headache, 4 cerebrovascular diseases, 5 psychiatric disorders, 7 optic neuritis). All control subjects underwent lumbar puncture as a part of the diagnostic workup, and the assessment of CSF AD core biomarkers yielded results within the normal range. Demographic and biochemical data of the cohort are shown in Table 1.

### 2.2. CSF Sampling and Analysis

Following standardized international guidelines [22], 10–12 mL of CSF was collected with sterile polypropylene tubes and centrifuged at room temperature at 2000× *g* for 10 min before storage; aliquots of 0.5 mL were stored at −80 °C. CSF AD core biomarkers (β-amyloid_1-40_ (Aβ40), β-amyloid_1-42_ (Aβ42), total tau (t-tau) and phosphorylated tau at threonine 181 (p-tau)) were measured in Perugia using a Lumipulse G600-II fully automated chemiluminescent enzyme immunoassay system (Fujirebio Europe, Gent, Belgium).

### 2.3. CSF Serpina1 Analysis by CIEF Immunoassay

Analysis of CSF serpinA1 was performed in Ulm (Centre for biomedical research, University of Ulm, Ulm, Germany) using a NanoPro 1000 CIEF immunoassay platform, reagents (pH gradients, capillaries and fluorescent standards) and Compass software from ProteinSimple (Santa Clara, CA, USA), as previously described [9]. Monoclonal mouse anti-human serpinA1 antibody was purchased from R&D System (MAB1268, Minneapolis, MN, USA). Samples were tested in duplicate in four 384-wells plates, and reproducibility was assessed by measuring the same sample in duplicate in all plates. The average coefficient of variation (CV) for intra- and inter-assay reproducibility was calculated as 13.6% and 17.9%, respectively.

Details about the CIEF method are available in our previous validation studies [9,10]. In brief, the output of CIEF analysis is an electropherogram with signal peaks indicating individual protein isoforms (see Figure 1). The software also provides values for absolute and relative peak areas that can be automatically exported and analyzed. In particular, the absolute areas (expressed in chemiluminescence units, CUs) represent the calculated area under each peak, whereas relative areas indicate the relative contribution of each peak (as a percentage) to the total absolute area computed by summing the individual areas of all peaks. First, we investigated the relative distribution of individual isoforms in terms of relative peak areas among the major diagnostic groups (AD, LBD and controls); then, we focused on the most significant isoforms for subsequent analysis among diagnostic subgroups (i.e., AD-MCI, AD-dem, PD, PD-MCI, PDD and DLB).

### 2.4. Statistical Analysis

Kruskal–Wallis (followed by Dunn–Bonferroni’s post hoc test) or χ^2^ tests were adopted to compare continuous (i.e., demographic and peak data) or categorical variables (i.e., sex) among groups, respectively. Spearman rho’s test was chosen to evaluate the correlations between the relative distribution of each isoform and AD core biomarkers. The differences in relative isoform expression were calculated by considering the mean values among groups. *p* < 0.05 was considered as the first level of statistical significance.

Finally, receiver operating characteristic (ROC) analysis was performed to assess the accuracy of peak analysis in terms of distinguishing the diagnostic groups. The maximized Youden’s index was used to calculate the best cutoff for each peak. All statistical analyses were performed using GraphPad Prism 7 (GraphPad software, La Jolla, CA, USA).

## 3. Results

### 3.1. Distribution of CSF SerpinA1 Isoforms

The typical CIEF electropherogram of a control subject consisted of six distinct isoforms with isoelectric points (pI) ranging from 4.3 to 4.7 (peaks 1–6) (Figure 1A). In a subset of samples, mainly LBD patients, a seventh isoform on the acidic side of the pattern was also observed (peak 0). The distribution pattern of serpinA1 isoforms but not the total absolute peak area differed in AD patients compared to controls (*p* < 0.0001) (Figure 1A,B and Figure 2A). A lower total absolute area (*p* < 0.02) and a different isoform expression were observed in AD with respect to LBD patients (Figure 1C and Figure 2A).

When comparing isoform expression patterns among groups, isoform 0 was almost undetectable in AD and controls and was found to be higher in LBD compared to AD patients (*p* < 0.0001) (Figure 2B). On the acidic side of the distribution pattern, isoform 1 was reduced in AD compared to LBD patients (−33.1%, *p* < 0.0001) (Figure 2C), whereas isoform 2 was decreased both with respect to controls (−28.6%, *p* = 0.0001) and LBD cases (−39.0%, *p* < 0.0001) (Figure 2D). The most abundant isoforms in the three groups were isoforms 3 and 4, which were increased in AD versus control subjects (+14.9%, *p* < 0.001 for isoform 4) and versus LBD patients (+8.4%, *p* < 0.01 for isoform 3; +23.1%, *p* < 0.0001 for isoform 4) (Figure 2E,F). On the basic side of the pattern, isoform 5 expression was higher in both AD and LBD patients versus controls (+46.2% and +29.6% respectively, *p* < 0.01 for both) (Figure 2G), whereas no differences were observed for isoform 6 (Figure 2H).

### 3.2. CSF SerpinA1 Isoforms in Diagnostic Subgroups

Figure 3 shows the relative abundance of isoforms 0, 1, 2 and 4 after further stratification of AD and LBD into diagnostic subgroups according to the disease stage. No significant differences in the relative abundance of any individual isoform were observed between AD-MCI and AD-dem. Isoforms 0 and 2 were significantly less expressed in AD-dem than in control subjects (−63.5% with *p* < 0.05 for peak 0; −33.4% and *p* < 0.02 for peak 2), PDD/DLB (−81.4% and −45.0% for peak 0 and 2, respectively; *p* < 0.0001 for both). Isoform 2 was also decreased in AD-MCI compared to both controls (−24.3%, *p* < 0.05) and PD-MCI (−36.9%, *p* < 0.0001). Conversely, isoform 4 was significantly increased both in AD-dem (+18.2% with *p* < 0.01 vs. controls; +27.5% with *p* < 0.0001 vs. PDD/DLB) and AD-MCI (+25.5% with *p* < 0.001 vs. PD-MCI).

### 3.3. Receiver Operating Characteristic (ROC) Analysis

ROC analysis (Table 2) evidenced that isoforms 2 and 4 had the highest discriminating potential both to differentiate AD in comparison to control subjects (area under the curve (AUC): 0.80 and 0.76, respectively) and to LBD (AUC: 0.92 and 0.85, respectively) (Figure 4). In the discrimination of AD patients from controls, the analysis of isoform 2 provided the highest sensitivity (83.6%), whereas that of isoform 4 showed the highest specificity (89.7%). Similarly, the same two isoforms achieved high performance in distinguishing AD from LBD (AUC relative to isoforms 2 and 4: 0.92 and 0.85, respectively), especially when considering only demented patients (AUC: 0.97 and 0.92 for isoforms 2 and 4, respectively) (Table 2). Both in the comparison of AD patients with controls and with LBD, the best cutoff for isoform 2 relative expression, as calculated by means of the maximized Youden’s index, was around 14.3%. A relative expression of isoform 2 lower than 14.3% was associated with a 4- and 24-fold higher risk of having AD versus controls and LBD, respectively (Table 2).

### 3.4. Correlations with AD Core Biomarkers

The expression of isoforms 0 to 4 showed significant correlations with the CSF biomarkers (Appendix A). In particular, the most acidic isoforms (peaks 0, 1 and 2) were positively correlated in the whole cohort with the Aβ42/40 ratio and negatively correlated both with p-tau and t-tau values (*p* < 0.001 for all). That is, a decrease in isoform 2 expression, as observed in AD, correlated with a lower Aβ42/40 ratio (Spearman rho (r) value: 0.48; 95% confidence interval (95% CI): 0.34 to 0.61) and higher p-tau (r: −0.50, 95% CI: −0.62 to −0.36) and t-tau levels (r: −0.50, 95% CI: −0.62 to −0.36) (*p* < 0.0001 for all). On the other hand, isoform 4 presented with an opposite trend, with a negative correlation with Aβ42/40 (r: −0.36, 95% CI: −0.51 to −0.20) and a positive correlation both with t-tau (r: 0.48, 95% CI: 0.32 to 0.59) and p-tau levels (r: 0.48, 95% CI: 0.33 to 0.60) (*p* < 0.0001 for all). This means that the altered expression of isoforms 2 and 4, which was specifically observed in AD patients, was associated with low Aβ42/40 and high p-tau and t-tau values.

When extending the analysis to diagnostic subgroups, we observed different correlations between serpinA1 isoforms and CSF biomarkers according to the disease stage (complete data are available in Appendix A). In demented patients with either AD or LBD, pathologic changes of serpinA1 isoform expression were more associated with CSF p-tau and t-tau levels. On the contrary, patients in the MCI stage showed stronger associations between serpinA1 isoforms and Aβ42/40 levels. We observed no difference between AD-MCI and AD-dem in terms of any of the three CSF AD core biomarkers.

## 4. Discussion

In recent years, serpinA1 has raised interest as a candidate fluid biomarker for neurodegenerative disorders, with its CSF and serum concentrations reported to be increased in DLB and AD [8]. The approach described in this work provided a further insight into the characterization of patients affected by neurodegenerative disorders, particularly of AD patients. We evidenced a specific pattern of serpinA1 isoforms in AD patients, which differs from that of LBD and control subjects but also from those previously described in CJD, FTLD and PDD [9,10]. Each neurodegenerative disorder investigated to date has shown a disease-specific pattern.

Overall, our data add value to the molecular characterization of neurodegenerative dementias. We found similarities between PDD and CJD (increased expression of the acidic peaks), as well as between AD and FTLD (decreased peak 2) [9,10]. On the one hand, these findings may indicate common pathological pathways involving serpinA1 in different proteinopathies. On the other hand, the finding of distinct spectra among different forms and subtypes of neurodegenerative disorders may suggest disease-specific alterations of the CSF serpinA1 pattern. Assessment of isoforms 2 and 4 was highly accurate in discriminating between AD-dem and PDD/DLB, with AUC values of 0.97 and 0.92, respectively, whereas isoform 0 was specifically linked to cognitively impaired LBD patients. Interestingly, isoform 4 was increased only in AD but not in other diagnostic groups. Moreover, additional specific alterations in FTLD and prion disease were observed only according to molecular subtypes, suggesting the specific involvement of individual misfolded proteins and/or strains in serpinA1 pathophysiology [10].

Considering the heterogeneity of these findings, it is reasonable to suggest that serpinA1 isoforms might be related to neurodegenerative disorders in a disease-specific fashion. The isoforms detected through CIEF immunoassay may indicate different post-translationally modified species of serpinA1, among which phosphorylation [9], glycosylation [23,24] and sialylation [7] are of critical importance in determining the correct folding, secretion and degradation of the protein. In systemic diseases involving serpinA1, such as serpinA1 deficiency [25], serpinA1 PTMs are being intensively investigated, but their role in neurodegeneration remains unclear, and it is still unknown which PTMs correspond to which serpinA1 isoforms. One hypothesis is that sialylated serpinA1 isoforms are represented by the acidic peaks of the CIEF electropherogram, according to previous results [9], and might play protective roles against excessive protein phosphorylation [7,26]. Within this framework, a major question is whether and how the diverse PTMs can impact the pathophysiological activities of serpinA1. The relative abundance of serpinA1 isoforms may affect its role of inhibiting and degrading proinflammatory proteases [1], leading to an impaired inflammatory response during neurodegeneration. Moreover, pathological PTMs may promote protein aggregation, explaining the consistent findings of misfolded and aggregated forms of serpinA1 reported in AD [6,27]. The fact that we found no differences in any of the detectable isoforms among the non-neurodegenerative neurological conditions included in the control group (data not shown) suggests that alteration of the serpinA1 isoform pattern might be a primary characteristic of neurodegenerative disorders. Targeting individual serpinA1 isoforms or PTMs showing disease-specific changes, hence, may represent an additional strategy against the pathological events experienced during neurodegeneration, such as protein aggregation and neuroinflammation. However, a better comprehension of all pathophysiological aspects of serpinA1 isoforms, especially in non-neuronal tissues is required before pursuing such directions.

Additional information provided by our study concerns the differential expression of serpinA1 isoforms in AD according to the disease stage. In particular, the altered expression of isoforms 1 and 4 was mainly observed in AD-dem patients, whereas that of isoforms 2 and 5 was found to a similar extent in AD-MCI and AD-dem. This may suggest that serpinA1 is involved in AD pathophysiology from early stages, also showing distinct features across the clinical continuum. Considering that these AD-related changes (in particular, the decrease in acidic isoforms and the increase in more basic isoforms) were associated with CSF levels of Aβ42/40, p-tau and t-tau, we might suppose that serpinA1 PTMs and AD neuropathological changes may relate to each other, even if in a small number of patients, we observed that the DLB subgroup showed mixed features that partly resembled those of LBD and AD patients. Although mainly speculative, our findings may suggest that a mixed LB and AD pathology, which is often reported in LBD patients [28,29], might affect the differential expression of serpinA1 isoforms.

Finally, assessment of serpinA1 isoforms through a CIEF immunoassay provided good diagnostic value in differentiating AD patients from controls and LBD patients, with maximal results in the discrimination between demented patients affected by either AD or LBD. Confirming our previous results that isoform 0 might be selectively involved in the development of dementia due to LBD [9], analysis of serpinA1 isoforms might represent an interesting diagnostic supplement, especially for patients with neurodegenerative dementias.

Overall, the strength of this serpinA1 CIEF immunoassay lies on the precise and reproducible detection of serpinA1 patterns by testing only a small amount of CSF (less than 10 µL), which allows for an accurate discrimination in a limited range of pH. On the other hand, the use of this technique in clinical practice is yet to be validated, and the need for quantitative rather than semi-quantitative data should be considered in the future. We acknowledge, as a main limitation of our study, the lack of a detailed characterization of serpinA1 individual isoform PTMs. However, we previously reported that sialylation is a relevant PTM present in PD and PDD patients [7]. Future studies with a CIEF assay should include experiments with an additional enzymatic pretreatment to confirm our previous findings. However, the identification of disease-specific isoforms of interest could allow for a more focused investigation. Moreover, we are aware of the small sample size of some patient subgroups, such as that of DLB patients; we intended this study to be a first exploratory step with respect to the role of serpinA1 in AD, and a deep characterization of LBD fell outside of our research scope.

In conclusion, we provided evidence of a specific distribution pattern of serpinA1 isoforms that allows to discriminate AD versus controls and LBD from early disease stages. Further studies with larger cohorts are required to better clarify the pathological link (e.g., cause or epiphenomenon) between serpinA1 isoforms and the molecular pathways underlying neuroinflammation and neurodegeneration.

## Figures and Tables

**Figure 1 ijms-23-06922-f001:**
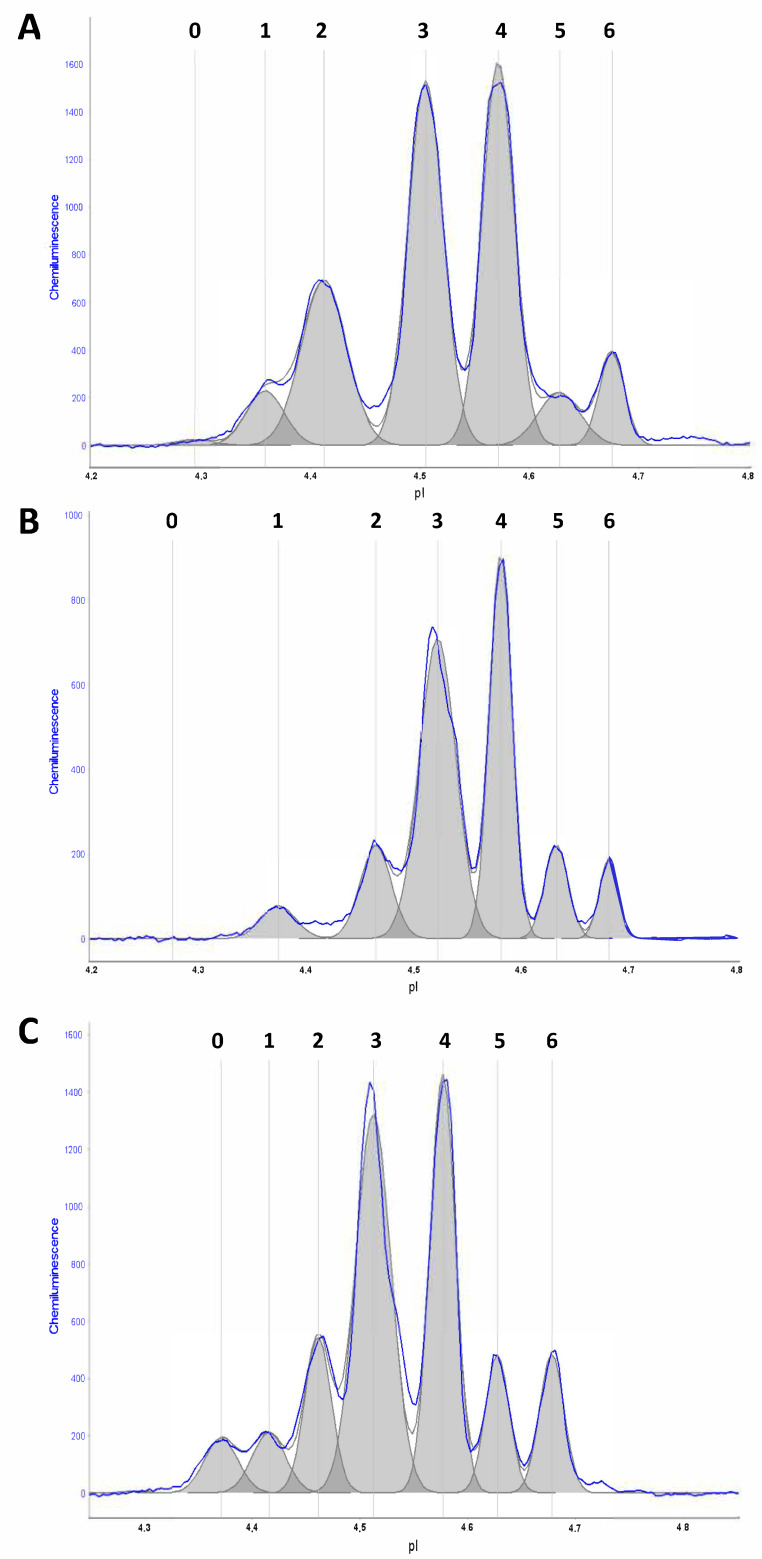
Typical CIEF electropherograms of serpinA1 isoforms. Exemplary CIEF electropherograms of (**A**) control subjects (six distinct peaks in the pH range of 4.3–4.7), (**B**) patients with Alzheimer’s disease with dementia (AD-dem) (with a relative decrease in isoform 2 and an increase in peak 4) and (**C**) patients with Parkinson’s disease with dementia (PDD). The grey color indicates the area under each peak. Signal intensity is reported in chemiluminescence units (CUs). pI: isoelectric point; exposure time: 120 s.

**Figure 2 ijms-23-06922-f002:**
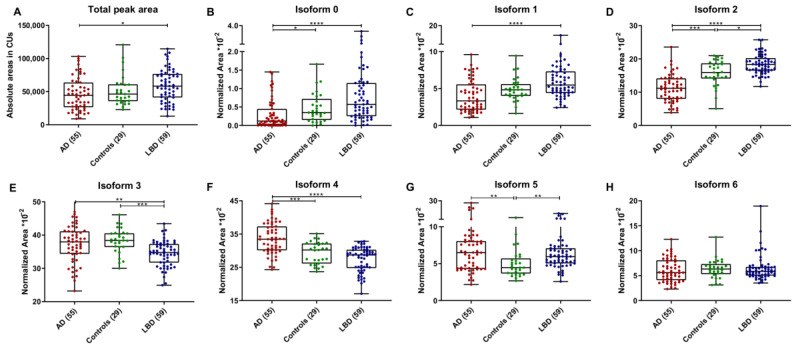
SerpinA1 isoform expression in AD, controls and LBD. (**A**) Comparison of total absolute peak areas among diagnostic groups (AD: Alzheimer’s disease, *n* = 55; Controls, *n* = 29; LBD: Lewy body disorders, *n* = 59), expressed in chemiluminescence units (CUs). (**B**–**H**) Relative expression of isoforms 0–6, reported as a percentage (normalized area *10^−2^). Box plots indicate median value, interquartile range and range of values. * *p* < 0.05, ** *p* < 0.01, *** *p* < 0.001, **** *p* < 0.0001.

**Figure 3 ijms-23-06922-f003:**
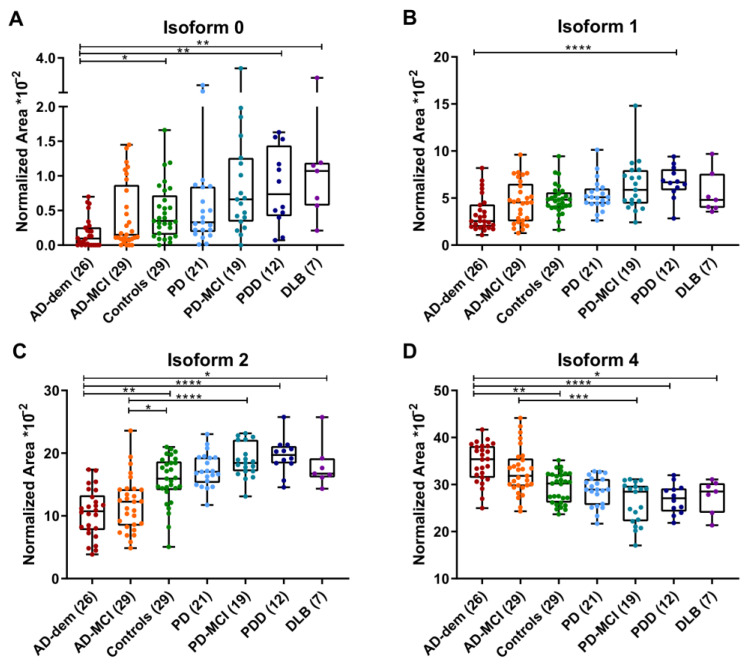
Relative expression of serpinA1 isoforms 0, 1, 2 and 4 among diagnostic subgroups. (**A**–**D**) Comparison of the relative expression of serpinA1 isoforms in controls, AD-MCI, AD-dem, PD, PD-MCI, PDD and DLB. Box plots indicate median value, interquartile range and range of values. * *p* < 0.05, ** *p* < 0.01, *** *p* < 0.001, **** *p* < 0.0001. Abbreviations. AD-dem: Alzheimer’s disease with dementia; AD-MCI: Alzheimer’s disease with mild cognitive impairment; DLB: dementia with Lewy bodies; PD: Parkinson’s disease; PDD: Parkinson’s disease with dementia; PD-MCI: Parkinson’s disease with mild cognitive impairment.

**Figure 4 ijms-23-06922-f004:**
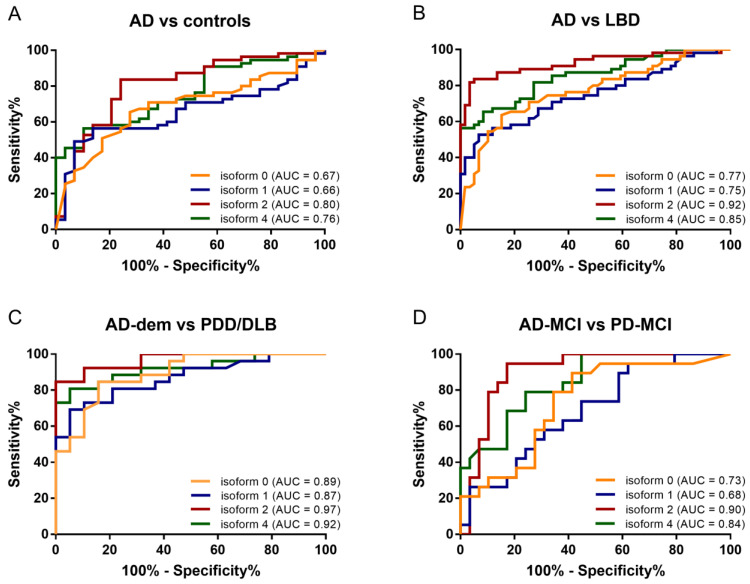
Receiver operating characteristic analysis. ROC curves relative to expression of isoforms 0, 1, 2 and 4 for discrimination of (**A**) AD vs. controls, (**B**) AD vs. LBD, (**C**) AD-dem vs. PDD/DLB and (**D**) AD-MCI vs. PD-MCI. Abbreviations. AD: Alzheimer’s disease; AD-dem: Alzheimer’s disease with dementia; AD-MCI: Alzheimer’s disease with mild cognitive impairment; AUC: area under the curve; DLB: dementia with Lewy bodies; LBD: Lewy body disease; PDD: Parkinson’s disease with dementia; PD-MCI: Parkinson’s disease with mild cognitive impairment; ROC: receiver operating characteristic.

**Table 1 ijms-23-06922-t001:** Demographic and biochemical data of the diagnostic groups.

	**AD**			**Controls**	**LBD**				
	AD-dem	AD-MCI	Total	Total	PD	PD-MCI	PDD	DLB	Total
N	26	29	55	29	21	19	12	7	59
Age *	70.7 ± 7.8	73.7 ± 4.7	72.2 ± 6.5	59.7 ± 13.2	61.2 ± 9.5	67.9 ± 7.4	71.3 ± 9.5	71.7 ± 9.5	66.7 ± 9.7
Females (%) **	17 (65.4%)	16 (55.2%)	33 (60.0%)	12 (41.4%)	7 (33.3%)	4 (21.1%)	1 (8.3%)	2 (28.6%)	14 (23.7%)
Aβ42/40 ratio	0.052 ± 0.010	0.049 ± 0.012	0.051 ± 0.011	0.114 ± 0.024 ***	0.105 ± 0.022	0.097 ± 0.030	0.079 ± 0.035	0.059 ± 0.019	0.095 ± 0.030
p-tau (pg/mL)	107.2 ± 54.2	99.5 ± 37.1	103.3 ± 46.2	44.1 ± 9.3 ***	37.3 ± 14.3	40.6 ± 22.1	43.8 ± 18.8	55.13 ± 33.0	40.5 ± 19.9
t-tau (pg/mL)	814.7 ± 324.3	756.8 ± 322.0	785.7 ± 321.7	261.9 ± 74.9 ***	248.4 ± 120.5	284.4 ± 172.0	274.2 ± 126.7	373.7 ± 188.0	273.8 ± 149.7
Total peak area (CUs)	42,791 ± 24,716	49,488 ± 24,654	46,322 ± 24,685	51,763 ± 23,350	50,894 ± 22,916	64,071 ± 23,332	64,759 ± 21,305	61,511 ± 33,844	59,217 ± 24,380
isoform 0 (%)	0.2 ± 0.2	0.4 ± 0.4	0.3 ± 0.3	0.5 ± 0.4	0.6 ± 0.6	0.9 ± 0.8	0.8 ± 0.8	1.1 ± 0.8	0.8 ± 0.7
isoform 1 (%)	3.3 ± 1.9	4.6 ± 2.2	4.0 ± 2.1	4.9 ± 1.6	5.3 ± 1.7	6.3 ± 2.8	6.7 ± 2.3	5.5 ± 2.3	6.0 ± 2.2
isoform 2 (%)	10.5 ± 3.8	11.9 ± 4.3	11.2 ± 4.1	15.7 ± 3.9	17.4 ± 2.7	18.8 ± 2.7	19.6 ± 3.7	18.1 ± 3.7	18.4 ± 2.9
isoform 3 (%)	37.2 ± 6.4	37.6 ± 5.2	37.4 ± 5.7	38.3 ± 3.7	35.8 ± 3.2	33.5 ± 3.4	32.6 ± 4.6	36.9 ± 4.6	34.5 ± 3.8
isoform 4 (%)	34.6 ± 4.2	32.8 ± 4.9	33.7 ± 4.7	29.3 ± 3.3	28.6 ± 3.1	26.1 ± 4.4	27.0 ± 3.6	27.5 ± 3.6	27.4 ± 3.7
isoform 5 (%)	8.3 ± 6.3	6.5 ± 3.3	7.3 ± 5.0	5.0 ± 2.0	6.4 ± 2.6	7.8 ± 3.2	5.2 ± 1.5	5.6 ± 1.5	6.5 ± 2.6
isoform 6 (%)	5.9 ± 2.2	6.2 ± 2.4	6.1 ± 2.3	6.3 ± 0.8	5.8 ± 1.0	6.6 ± 2.3	8.2 ± 0.9	5.3 ± 0.9	6.5 ± 2.6

Data are reported as mean ± standard deviation. Total peak area and relative expression of serpinA1 isoforms are expressed in chemiluminescence units (CUs) and as a percentage, respectively. * Significant difference between AD and controls (*p* < 0.0001), as well as AD and LBD (*p* < 0.01). ** Significant difference between AD and LBD (*p* < 0.0001). *** CSF biomarker levels available for 20 patients. Abbreviations. Aβ42/40: amyloid-β_1-42_/amyloid-β_1-40_ ratio; AD: Alzheimer’s disease; AD-dem: Alzheimer’s disease with dementia; AD-MCI: Alzheimer’s disease with mild cognitive impairment; CSF: cerebrospinal fluid; DLB: dementia with Lewy bodies; LBD: Lewy body disease; PD: Parkinson’s disease; PDD: Parkinson’s disease with dementia; PD-MCI: Parkinson’s disease with mild cognitive impairment; p-tau: phosphorylated tau protein at threonine 181; t-tau: total tau protein.

**Table 2 ijms-23-06922-t002:** Diagnostic accuracy of serpinA1 isoform analysis in Alzheimer’s disease.

	**Mean Variation%**	**AUC (95% CI)**	**Best Cutoffs**	**Sensitivity% (95% CI)**	**Specificity% (95% CI)**	**LR+**
	AD vs. Controls					
isoform 0	−36.6 *	0.67 (0.57–0.80)	0.2	65.5 (51.4–77.8)	72.4 (52.8–87.3)	2.4
isoform 1	−	0.66 (0.54–0.77)	3.8	56.4 (42.3–69.7)	86.2 (68.3–96.1)	4.1
isoform 2	−28.6 ***	0.80 (0.70–0.91)	14.4	83.6 (71.2–92.2)	75.9 (56.5–89.7)	3.5
isoform 3	−	0.54 (0.42–0.66)	36.4	40.0 (27.0–54.1)	79.3 (60.3–92.0)	1.9
isoform 4	+14.9 ***	0.76 (0.66–0.86)	33.0	56.4 (42.3–69.7)	89.7 (72.7–97.8)	5.5
isoform 5	+46.2 **	0.69 (0.57–0.81)	6.2	54.6 (40.6–68.0)	86.2 (68.3–96.1)	4.0
isoform 6	−	0.55 (0.42–0.67)	5.5	49.1 (35.4–62.9)	72.4 (52.8–87.3)	1.8
	AD vs. LBD					
isoform 0	−62.1 ****	0.77 (0.65–0.86)	0.2	63.6 (49.6–76.2)	84.8 (73.0–92.8)	4.2
isoform 1	−33.1 ****	0.75 (0.66–0.84)	3.5	52.7 (38.8–66.4)	93.2 (83.5–98.1)	7.8
isoform 2	−39.0 ****	0.92 (0.87–0.98)	14.3	81.8 (69.1–90.9)	96.6 (88.3–99.6)	24.1
isoform 3	+8.4 **	0.68 (0.58–0.78)	38.6	45.5 (32.0–59.5)	91.5 (81.3–97.2)	5.4
isoform 4	+23.1 ****	0.85 (0.78–0.92)	31.4	65.5 (51.4–77.8)	91.5 (81.3–97.2)	7.7
isoform 5	+12.8 **	0.52 (0.41–0.63)	7.5	41.8 (28.7–55.9)	84.8 (73.0–92.8)	2.7
isoform 6	−	0.55 (0.44–0.66)	4.2	27.3 (16.2–41.0)	94.9 (85.9–98.9)	5.4
	AD-dem vs. PDD/DLB					
isoform 0	−81.4 ****	0.89 (0.80–0.99)	0.4	84.6 (65.1–95.6)	84.2 (60.4–96.6)	5.4
isoform 1	−47.0 ****	0.87 (0.76–0.97)	3.6	69.2 (48.2–85.7)	94.7 (74.0–99.9)	13.2
isoform 2	−45.0 ****	0.97 (0.93–1.00)	14.3	84.6 (65.1–95.6)	99.9 (82.4–100.0)	84.6
isoform 3	−	0.68 (0.52–0.84)	38.5	50.0 (29.9–70.1)	89.4 (66.9–98.7)	4.8
isoform 4	+27.5 ****	0.92 (0.84–0.99)	31.1	80.8 (60.7–93.5)	94.7 (74.0–99.9)	15.4
isoform 5	−	0.67 (0.52–0.83)	7.5	46.2 (26.6–66.6)	99.9 (82.4–100)	84.6
isoform 6	−	0.58 (0.41–0.75)	10.1	96.2 (80.4–99.9)	26.3 (9.1–51.2)	1.3
	AD-MCI vs. PD-MCI					
isoform 0	−	0.73 (0.59–0.88)	0.2	89.5 (66.9–98.7)	58.6 (38.9–76.5)	2.2
isoform 1	−	0.68 (0.53–0.83)	3.5	94.7 (74.0–99.9)	37.9 (20.7–57.7)	1.5
isoform 2	−36.9 ****	0.90 (0.81–0.99)	14.4	94.7 (74.0–99.9)	79.3 (60.3–92.0)	4.6
isoform 3	+12.5 *	0.75 (0.62–0.89)	37.9	100.0 (82.4–100.0)	48.3 (29.5–67.5)	1.9
isoform 4	+25.5 ***	0.84 (0.74–0.95)	31.3	100.0 (82.4–100.0)	55.2 (35.7–73.6)	2.2
isoform 5	−	0.65 (0.49–0.80)	4.6	100.0 (82.4–100.0)	37.9 (20.7–57.7)	1.6
isoform 6	−	0.54 (0.38–0.71)	4.1	100.0 (82.4–100.0)	24.1 (10.3–43.5)	1.3

Analysis of the relative expression of isoforms 0–6 in AD patients versus controls and LBD. Mean variation is reported as percentage (*p*-value). Data concerning the mean variation are reported only when statistically significant. Sensitivity, specificity and positive likelihood ratio (LR+) values refer to the cutoffs calculated by maximizing the Youden’s index. The 95% confidence interval (95% CI) for sensitivity and specificity are presented in parentheses. * *p* < 0.05, ** *p* < 0.01, *** *p* < 0.001, **** *p* < 0.0001. Abbreviations. AD: Alzheimer’s disease; AD-dem: Alzheimer’s disease with dementia; AD-MCI: Alzheimer’s disease with mild cognitive impairment; AUC: area under the curve; DLB: dementia with Lewy bodies; LBD: Lewy body disease; PDD: Parkinson’s disease with dementia; PD-MCI: Parkinson’s disease with mild cognitive impairment.

## Data Availability

The data that support the findings of this study are available from the corresponding author upon reasonable request.

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
