# Peer review of "Specific Cerebrospinal Fluid SerpinA1 Isoform Pattern in Alzheimer’s Disease"

_ijms, 2022, doi:10.3390/ijms23136922_

Round 1
Reviewer 1 Report
SerpinA1 is an acute inflammatory protein, which seems to play a role in neurodegeneration and neuroinflammation. In Alzheimer's disease and synucleinopathies, SerpinA1 is overexpressed in the brain and the cerebrospinal fluid (CSF) showing abnormal patterns of its charge isoforms. To date, no comprehensive studies explored SerpinA1 CSF isoforms in AD. Barba et al. reported their interesting findings, but there are some gaps that need to be addressed further.
1. Introduction: SerpinA1 CSF isoforms in Creutzfeldt-Jakob disease (CJD) and frontotemporal lobar degeneration (FTLD) was also found, this needs to be mentioned
2. Demographics and CSF biomarker results in the diagnostic groups that need to summary in a table
3. Among 6 isoforms, the authors need to show which is value for each group and discuss further outcomes rather than just presenting in tables
4. Since SerpinA1 isoforms were repeatedly reported in Neurological disorders, could be common features of most neurodegenerative disorders?
Author Response
Reviewer 1
SerpinA1 is an acute inflammatory protein, which seems to play a role in neurodegeneration and neuroinflammation. In Alzheimer's disease and synucleinopathies, SerpinA1 is overexpressed in the brain and the cerebrospinal fluid (CSF) showing abnormal patterns of its charge isoforms. To date, no comprehensive studies explored SerpinA1 CSF isoforms in AD. Barba et al. reported their interesting findings, but there are some gaps that need to be addressed further.
We thank the Reviewer for the positive comments and for the suggestions.
- Introduction: SerpinA1 CSF isoforms in Creutzfeldt-Jakob disease (CJD) and frontotemporal lobar degeneration (FTLD) was also found, this needs to be mentioned.
We agree with the Reviewer and, accordingly, we addressed this point in the Introduction section, as follows (see lines 51-58):
“Moreover, the qualitative assessment of serpinA1 has gained increasing attention in the last few years, since different expression patterns of serpinA1 isoforms have been described in several neurodegenerative disorders. The first report on PD with dementia (PDD) described a shift towards more acidic pH values and the presence of an additional acidic isoform in comparison to PD without dementia (9). Subsequently, the same approach was applied to investigate patients with Creutzfeldt-Jakob disease (CJD) and fronto-temporal lobar degeneration (FTLD), who showed peculiar patterns with higher expression of acidic and basic isoforms, respectively (10). So far, no study focused on serpinA1 isoforms and their relevance in AD.”
- Demographics and CSF biomarker results in the diagnostic groups that need to summary in a table
Accordingly to the Reviewer’s suggestion, we improved the Table 1 with additional data about results analyses.
- Among 6 isoforms, the authors need to show which is value for each group and discuss further outcomes rather than just presenting in tables
We are thankful for the suggestion. We addressed the issue in the Discussion section, as follows (see lines 206-216):
“We found similarities between PDD and CJD (increased expression of the acidic peaks) as well as between AD and FTLD (decreased peak 2) (9,10). On the one hand, these findings may indicate common pathological pathways involving serpinA1 in different proteinopathies. On the other hand, the finding of distinct spectra among different forms and subtypes of neurodegenerative disorders may suggest disease-specific alterations of CSF SerpinA1 pattern. The assessment of isoforms 2 and 4, indeed, provided high accuracy in the discrimination between AD-dem and PDD/DLB, with AUC values of 0.97 and 0.92, respectively, while isoform 0 was specifically linked to cognitively impaired LBD patients. Interestingly, isoform 4 was increased only in AD but not in other diagnostic groups. Moreover, additional specific alterations in FTLD and prion disease could be observed only after stratification for molecular subtypes, suggesting the specific involvement of individual misfolded proteins and/or strains in serpinA1 pathophysiology (10).”
- Since SerpinA1 isoforms were repeatedly reported in Neurological disorders, could be common features of most neurodegenerative disorders?
Exactly, and we found interesting that no alterations could be observed in non-degenerative neurological disorders, letting us suppose the specific involvement of serpinA1 in the neurodegenerative process. We discussed the issue in the last section as follows (see lines 230-235):
“Moreover, pathological PTMs may promote protein aggregation, thus explaining the consistent findings of misfolded and aggregated forms of serpinA1 reported in AD (6,27). The fact that we found no differences in any of the detectable isoforms among the non-neurodegenerative neurological conditions included in the control group (data not shown) suggests that alterations of serpinA1 isoform pattern might be a primary characteristic of neurodegenerative disorders.”

Reviewer 2 Report
This manuscript introduces CSF serpinA1 isoform patterns as the potent biomarkers of AD. The writing is straightforward and it attracted to the readership of the International Journal of Molecular Sciences; however, major revision of the manuscript is necessary prior to publication.
Comments:
1. In Figure 1, the “peak 0” in Figure 1A and 1B are very tiny peak. Can they be considered as peak?
2. In page 4, first line, the authors mentioned that “the most abundant isoform in the three groups was isoform 3”. Based on the Figure 1, however, isoform 4 looks the most abundant one. Please double check this.
3. The authors compare the serpinA1 isoform patterns in AD, PD, and LBD patients to control. Please define “control” in the manuscript.
4. Please provide more information how the authors differentiate AD, PD, and LBD patients.
Author Response
Reviewer 2
This manuscript introduces CSF serpinA1 isoform patterns as the potent biomarkers of AD. The writing is straightforward and it attracted to the readership of the International Journal of Molecular Sciences; however, major revision of the manuscript is necessary prior to publication.
We are thankful to the Reviewer for the suggestions.
Comments:
- In Figure 1, the “peak 0” in Figure 1A and 1B are very tiny peak. Can they be considered as peak?
We are grateful to the Reviewer for the observation. “Peak 0” was undetectable in controls and AD patients of the present cohort, but also in FTLD and CJD patients, according to our previous publication (Abu-Rumeileh and Halbgebauer et al. 2020). “Peak 0” appears to be clearly quantifiable only in a subgroup of LBD patients, mainly PDD. Hence, we considered the additional peak as a biochemical correlate of cognitive impairment in LBD (Jesse et al. 2012, Halbgebauer et al. 2016). Here, we further observed that “peak 0” be useful in the differential diagnosis between AD and PDD/DLB. We tried to explain better the current issue in the Results and Discussion sections, as follows (see lines 144-146 and 210-213):
“When comparing isoform expression patterns among groups, isoform 0 was almost undetectable in AD and controls and was found higher in LBD (p<0.0001 vs AD) (Figure 2B).”
“The assessment of isoforms 2 and 4, indeed, provided high accuracy in the discrimination between AD-dem and PDD/DLB, with AUC values of 0.97 and 0.92, respectively, while isoform 0 was specifically linked to cognitively impaired LBD patients.”
- In page 4, first line, the authors mentioned that “the most abundant isoform in the three groups was isoform 3”. Based on the Figure 1, however, isoform 4 looks the most abundant one. Please double check this.
We agree with the Referee. In accordance with another comment on Figure 1, we selected a better explicative image to represent serpinA1 pattern in AD and LBD and modified accordingly the format of the panels. Since isoforms 3 and 4 were similar in most cases (sometimes isoform 4 was even more abundant than isoform 3), we corrected the text (see lines 148-150):
“The most abundant isoforms in the three groups were isoforms 3 and 4, which were increased in AD versus controls (+14.9%, p<0.001 for isoform 4) and versus LBD (+8.4%, p<0.01 for isoform 3; +23.1%, p<0.0001 for isoform 4)”.
- The authors compare the serpinA1 isoform patterns in AD, PD, and LBD patients to control. Please define “control” in the manuscript.
We have already reported the clinical characteristics of the control group in the Methods section of the previous draft, as follows (see lines 90-94):
“The control group (n = 29) was composed by subjects referring to the same Section of Neurology (University of Perugia, Perugia, Italy) for subjective memory complaint, who did not fulfill criteria for MCI at the neuropsychological evaluation (n = 12), and for non-degenerative neurological conditions (1 headache, 4 cerebrovascular diseases, 5 psychiatric disorders, 7 neuritis optica).”
- Please provide more information how the authors differentiate AD, PD, and LBD patients.
If the Reviewer refers to the diagnostic classification, please note that we grouped Parkinson’s disease (PD) and dementia with Lewy bodies (DLB) into one unique group (Lewy body disease, LBD) (Irwin et al. 2017), as mentioned in the Introduction:
“Hence, we took advantage of our previously described capillary isoelectric focusing (CIEF) immunoassay (9) to investigate qualitative modifications of CSF serpinA1 pattern in patients with AD, patients with Lewy body disease (LBD, including PD and DLB patients) and control subjects.”
If the comment refers, instead, to the CSF findings among the above mentioned groups, we have addressed the topic in the Discussion, as follows (see lines 210-213, the same as for the previous comment):
“The assessment of isoforms 2 and 4, indeed, provided high accuracy in the discrimination between AD-dem and PDD/DLB, with AUC values of 0.97 and 0.92, respectively, while isoform 0 was specifically linked to cognitively impaired LBD patients. Interestingly, isoform 4 was increased only in AD but not in other diagnostic groups”.

Reviewer 3 Report
The article by Lorenzo Barba and colleagues entitled "Specific cerebrospinal fluid Serpin A1 isoform patterns in Alzheimer´s disease" provides evidence for the presence of distinct proteoforms of SerpinA1 in a valuable collective of cerebrospinal fluid (CSF) from AD patients. Data presented suggest that SerpinA1-proteoforms/isoforms might exert distinct pathophysiologic roles in AD which differ from LBD, CJD, FTLD and PDD patient cohorts as reported earlier by the group of Markus Otto. Although it is still unknown which factors influence the differential SerpinA1-isoform abundances and which PTMs exactly might lead to the observed pI-dependent separation in electropherograms, this work add another piece of clinical evidence for a biological relevance of serpinA1 proteoforms and thus, I support a publication after consideration of some suggestions:
Minor comments: ad Fig 1) lettering size too small
Outreaching suggestions:
The authors showed in a previous publication that sialylation on SerpinA1 is a relevant PTM present in PD and PDD patients (PMID: 23144969). The 2D-DIGE- and selective MS/MS methodology used in this work, in combination with enzymatic removal of PTMs (in that case neuraminidase-treatment) is a representative example for the superior value of the 2D-DIGE methodology over nowadays commonly used shotgun proteomic workflows. Even though I am aware of the experimental efforts which are needed to do such an analysis on clinical cohorts, analysis of a subset of i.e. AD-patients vs controls with 2D-DIGE and differential treatment with enzymes such as neuraminidases, glucosidases, phosphatases would potentially bring even more insight into mechanisms involved or leading to SerpinA1-proteomforms in neurodegenerative disease, in this case AD.
Author Response
Reviewer 3
The article by Lorenzo Barba and colleagues entitled "Specific cerebrospinal fluid Serpin A1 isoform patterns in Alzheimer´s disease" provides evidence for the presence of distinct proteoforms of SerpinA1 in a valuable collective of cerebrospinal fluid (CSF) from AD patients. Data presented suggest that SerpinA1-proteoforms/isoforms might exert distinct pathophysiologic roles in AD which differ from LBD, CJD, FTLD and PDD patient cohorts as reported earlier by the group of Markus Otto. Although it is still unknown which factors influence the differential SerpinA1-isoform abundances and which PTMs exactly might lead to the observed pI-dependent separation in electropherograms, this work add another piece of clinical evidence for a biological relevance of serpinA1 proteoforms and thus, I support a publication after consideration of some suggestions:
Minor comments: ad Fig 1) lettering size too small
We thank the Reviewer for the positive comments and suggestions (Figure 1 has been revised).
Outreaching suggestions:
The authors showed in a previous publication that sialylation on SerpinA1 is a relevant PTM present in PD and PDD patients (PMID: 23144969). The 2D-DIGE- and selective MS/MS methodology used in this work, in combination with enzymatic removal of PTMs (in that case neuraminidase-treatment) is a representative example for the superior value of the 2D-DIGE methodology over nowadays commonly used shotgun proteomic workflows. Even though I am aware of the experimental efforts which are needed to do such an analysis on clinical cohorts, analysis of a subset of i.e. AD-patients vs controls with 2D-DIGE and differential treatment with enzymes such as neuraminidases, glucosidases, phosphatases would potentially bring even more insight into mechanisms involved or leading to SerpinA1-proteomforms in neurodegenerative disease, in this case AD.
We are thankful to the Reviewer for the precious suggestion and admit that it was considered as a part of the experimental design of this study. However, as the Reviewer mentioned, this is not feasible in a large clinical cohort. Apart from the technical considerations, the CSF volume of well characterized cohorts which is necessary for such a study (2D-PAGE, with different enzymes) is the major limitation. Such a study could only have a pilot character. We totally agree with the Reviewer on the potential added value of such findings, especially regarding sialylation patterns, since other PTMs have been found to be less detrimental in serpinA1 pathophysiology. Moreover, it would be surely of interest to replicate in future studies the enzymatic pre-treatment experiments with the CIEF assay, in order to evaluate if this method is sensitive enough to detect such qualitative differences. And yes, we aim to do this in a future study, also combining different enzymes. Accordingly, we discussed the above mentioned issue in the Discussion as a potential limiation of the paper, as follows (see lines 263-265):
“We acknowledge as a main limitation of our study the lack of a detailed characterization of serpinA1 individual isoform PTMs. However, we previously reported that sialylation is a relevant PTM present in PD and PDD patients (7). Future studies with the CIEF assay would greatly benefit of experiments with an additional enzymatic pre-treatment to confirm our previous findings.”

Round 2
Reviewer 1 Report
Thanks for revising thoroughly.
I have no further remarks, the manuscript is now improved significantly